# Investigation of Electrical Transitions in the First Steps of Spark Plasma Sintering: Effects of Pre-Oxidation and Mechanical Loading within Copper Granular Media

**DOI:** 10.3390/ma15124096

**Published:** 2022-06-09

**Authors:** Anis Aliouat, Guy Antou, Vincent Rat, Nicolas Pradeilles, Pierre -Marie Geffroy, Alexandre Maître

**Affiliations:** Institute for Research on Ceramics (IRCER), UMR CNRS 7315, University Limoges, F-87068 Limoges, France; anis.aliouat@unilim.fr (A.A.); vincent.rat@unilim.fr (V.R.); nicolas.pradeilles@unilim.fr (N.P.); pierre-marie.geffroy@unilim.fr (P.-M.G.); alexandre.maitre@unilim.fr (A.M.)

**Keywords:** granular medium, copper, electrical behavior, Branly effect, microstructure-properties correlation, Spark Plasma Sintering

## Abstract

Spark Plasma Sintering (SPS) has become a conventional and promising sintering method for powder consolidation. This study aims to well understand the mechanisms of densification encountered during SPS treatments, especially in the early stages of sintering. The direct current (DC) electrical behavior of copper granular medium is characterized. Their properties are correlated with their microstructural evolutions through post-mortem scanning electron microscope (SEM) observations to allow a thorough understanding of the involved Branly effect that is suspected to occur in SPS. The electrical response is studied by modifying the initial thickness of the oxide layer on particles surfaces and applying various mechanical loads on the granular medium. Without load and at low current, the measured quasi-reversible behavior is connected to the formation of spots at the microcontacts between the particles. By increasing the current, the Branly transition from an insulating to a conductive state suddenly occurs. The insulating oxide layer is destroyed, and micro-bridges are created. The application of a mechanical pressure strongly modifies the DC Branly effect. Increasing low stress leads to a strong decrease in the breakdown field. For high-applied pressure, successive drops in the electric field are detected during the electrical transition. These successive drops are induced by microcracking of the insulating oxide layer.

## 1. Introduction

Spark Plasma Sintering (SPS), which belongs to a set called the Field-Assisted Sintering Technique (FAST) [1,2], is used for the synthesis, consolidation and assembly of metals [3,4,5], ceramics [6,7] and composites materials [8,9,10]. This technique allows elaborating fully dense materials while maintaining the fine microstructure or even nanostructure of the starting powders [11,12,13,14,15]. This limitation of grain growth leads usually to improved in-service properties of SPSed specimens. These benefits are obtained thanks to the simultaneous application of a uniaxial stress (10–100 MPa) and a high heating rate (up to 1000 °C/min) induced by the flow of a pulsed electric current through the graphite die [16,17] or even within the granular medium (if the latter is conductive) [18,19].

However, the whole of the physical phenomena promoted by the combination of high-pulsed electric current and mechanical loading has not been clearly understood so far [20,21,22,23,24]. Many questions about the electrical, thermal and mechanical mechanisms involved during SPS treatment remain unanswered. In fact, the application of a pulsed current of high intensity leading to high heating rates could under certain conditions bring about specific processes, such as activation of thermal diffusion involved by possible high local temperature gradients [25], reduction in surface diffusion phenomenon resulting in grain growth at the beginning of the consolidation process [26] or electromigration (i.e., diffusion motion of the metallic ions in a metal under the effect of an electric field during the SPS process) [27,28,29,30,31]. This can substantially amend the materials behavior during SPS. Over the last few years, copper has been the main focus of much research with the objective of fully understanding the mechanisms occurring in the SPS process [32,33]. Copper is well-known for its remarkable electrical properties with high ductility and known sinterability [34]. For these reasons, it is the desired candidate to investigate the sintering mechanisms of conductive powders [3]. The effect of electric current on the sintering of Cu spheres (3 mm diameter) arranged on copper plates (1 mm thick) was studied by Frei et al. [35]. The experiment was performed by varying the current (0, 750, 850 and 1040 A) at a constant sintering temperature of 900 °C during 60 min. They showed that the current had a marked effect on the growth of the necks between the particles and the plates. In fact, with the presence of the highest current (1040 A), the neck size formed is almost five times greater than that resulting from sintering in the absence of current under the same conditions. They suggested that this sintering enhancement under the effect of current was attributed to electromigration. Some papers have investigated the occurrence of a local overheating above the melting point taking place at contact areas between particles; Yanagisawa et al. [36] have observed neck morphologies at the interparticle contacts after SPS sintering on copper powder of uniform size. It was found that Cu particles are joined together by melting, and necks are formed at points of contact between particles. These results suggest that melting and/or vaporization occur because of extremely high temperatures attained by the local heat generation at the interparticle contacts in the initial stage of sintering, and it easily exceeds 10,000 K when the contact surface and the relative density are low. However, Aman et al. [37] suggest mass transport by an ejection mechanism following the formation of unconventional necks on finer copper particles in the early stages of pressureless SPS. Zhang et al. [13] also observed similar non-conventional melted bridges between Cu particles. According to them, the interparticle necks develop thanks to a process of evaporation–condensation following the splashing of molten copper between the surfaces of the particles. In this regard, Song et al. [38] studied the mechanism of neck formation through a process of local melting and rapid solidification by using pure spherical Cu powders as the conducting sintering material. Based on their experimental findings, there was a considerable inhomogeneous distribution of temperature from the particle-contacting surface toward the core, in such a way that the maximum temperature is observed on the surface of the particle (i.e., the contact region of two particles). They concluded that the temperature on the contact region of the particles can reach 3000 °C, which is higher than the boiling point of Cu. Therefore, the surface areas of the Cu particles partially melt or evaporate, which leads to the neck formation. On the other hand, other studies suggest dielectric breakdown as a mechanism for cleaning the particle surfaces during SPS sintering [39,40]. Wu et al. [41] highlighted a surface-cleaning mechanism during the FAST process of Cu particles. Based on their results, the oxide layer is removed from the surface of Cu powders; the microstructural characterizations by TEM (Transmission Electronic Microscopy) show that sintering necks were formed after the oxides migration. They suggest that the dominant mechanism for the cleaning of surface is the localized high temperature produced by the instantaneous electric field concentration. Their results show that the pulsed electrical current affects strongly the sintering mechanism. At the same time, the role of applied electromagnetic (EM) fields is usually narrowed down to the generation of Joule heat at the macroscopic scale. Only a few works deal with the specific effects induced by these EM fields, which can significantly contribute to the enhancement of the densification process [27,35,42,43,44].

One of these suggested EM effects refers to the work carried out by Branly in 1890 [45] on the detection of waves and the spectacular “Branly effect”. Branly highlighted the existence of an electrical instability that appears in a metallic powder, slightly oxidized on the surface, under the application of a spark generated nearby. This effect corresponds to the transition of the granular medium from an insulating to a conductive state, characterized by the decrease in electrical resistance over several orders of magnitude, as soon as an electromagnetic wave is produced in its vicinity [46,47]. This EM interaction is supposed to create well-conductive electrical paths (micro-weldings) between the metallic particles and cause the electrical transition. Guyot et al. [44] compared the morphology of a copper granular medium submitted to a spark in its vicinity to the one observed during interrupted SPS treatments. In both cases, they observed that some particles showed a local melting, and others seemed to have been sprayed with the appearance of liquid copper. Following these observations, they suggested that EM effects would induce local micro-welds during the early stages of SPS sintering and thus promote SPS densification mechanisms.

A similar mechanism of conduction transition in granular medium, without any external excitation, is also observed when a DC current is injected through the sample and exceeds a certain value, which is usually considered as “a threshold voltage”. This phenomenon is called the DC Branly effect [48]. Falcon et al. [49] reported the observation of electrical transport through a chain (1D) of 50 identical stainless-steel beads with a diameter of 8 mm, covered by a thin layer of oxide. A static load between 10 and 500 N was applied. When the applied DC current is increased, a saturation voltage is reached. This saturation voltage is independent of the applied force, and it depends only on the number of beads between the electrodes. During this saturation, a “smooth” transition from an insulating to a conductive state is noticed. This transition is reversible; the initial resistance of the 1D chain can be restored by a shock on the tube containing the beads [49]. This smooth and reversible transition comes from an electro-thermal coupling in the vicinity of the microcontacts between each bead [50]. When the current is high enough, they supposed that electro-thermal processes occurred at the microcontacts, leading to high local temperature and destruction of the oxide layer on the surfaces of stainless-steel particles. Well-conducting electrical paths would thus be created [50]. In another study, Falcon et al. [51] reported a comparison of electrical conductivity measurements in two geometric configurations of granular systems using the same stainless steel beads: the first one in a 2D triangular lattice of metallic beads, and the second one in a 1D chain of beads, both submitted to mechanical loads. They noticed that the voltage–current (*U*-*I*) characteristics are qualitatively similar during both experiments, with a voltage saturation leading to a smooth electrical transition. For an isotropic compression of the 2D granular medium, an experimental map of the current paths has shown that the current paths were localized in few discrete linear paths, which explains the similarity between the electrical characteristics in the 1D and 2D lattices.

In the literature, only few works deal with the analysis of the electrical behavior of granular media in a real 3D configuration. Two works were conducted by Falcon et al. [52,53] on a granular medium composed of copper particles, with a mean diameter around 100 µm. A low uniaxial stress was applied, up to 20 MPa. As the voltage increased, the resistance–voltage characteristic (*R*-*U*) becomes non-linear. Beyond a critical value *(Uc)*, the voltage and resistance drop abruptly by several orders of magnitude, corresponding to a “sudden” transition from an insulating to a conductive state. Falcon et al. deduced that the major difference with the 1D experiment is that the critical voltage (*Uc*) in the 3D system depends on the force applied to the powder and that the voltage does not saturate but falls directly in an abrupt way.

Dorbolo et al. have also studied the electrical properties of 2D [54] and 3D [55,56] granular systems. Indeed, in this last study [56], the authors characterized a granular medium based on spherical lead beads with an average diameter around ≈2.35 mm. Their surface is composed of a nanometric layer of lead oxide. By injecting the current, the electrical resistance of the packing drops suddenly by more than two orders of magnitude. They suggested that this transition comes from a dielectric breakdown, with an avalanche of contacts breakdowns starting from the weakest barrier. They proposed that this dielectric breakdown would correspond to micro-soldering of the beads by a destruction of the oxide layer present on the surface due to the local temperature increase, which leads to micro-welding of the beads [47].

Recently, several experimental studies have been performed on the electrical properties of packing of metallic disks [57,58,59]. Their voltage–current characteristics present a static bearing and describe a hysteresis. Zorica et al. [59] suggested that electro-mechanical processes allow both the creation and breakdown of microcontacts between two stacked disks. According to them, the current would flow through contact spots related to the roughness of facing surfaces. Finally, recent experiments have been achieved on the electrical properties of metallic granular systems [60,61,62] in order to improve the understanding of the Branly effect. In these studies, voltage saturates with increasing current. The authors suggested that this saturation is due to the asperities of the rough surfaces of beads, resulting in many microcontacts and local welding.

All these previous studies have shown that the electrical conduction mechanisms within metallic granular media are complex. They suggested that the observed “smooth” or “sudden” electrical transitions would be related to electro-thermal processes occurring at interparticle microcontacts, which leads to microwelds. However, no microstructural evidence of these microcontacts formation has been provided. Moreover, the electro-thermal processes at the origin of the formation of interparticle microwelds as well as their chronology remain without a perfectly satisfactory understanding.

Therefore, the purpose of the present work is to approach the electrical response within granular metallic medium submitted to a DC current in a 3D configuration at micrometric scale. Thus, the electrical behavior (voltage–current characteristic) of a powder bed composed of pre-oxidized copper particles is carefully characterized. Different initial thicknesses of the oxide layer on the surfaces of the copper particles are obtained through controlled reduction–oxidation treatments. The influence of the pre-oxidation state on the electrical response is analyzed. In addition, electrical characterizations are carried out under low and high applied uniaxial stresses (up to 300 MPa) in order to analyze and attempt to establish correlations between the morphological and microstructural characteristics of the 3D granular medium (i.e., compactness, number and radius of contacts between the particles, etc.) and their electrical response. The microstructural modifications and their evolution induced by the flow of current are finely characterized by electron microscopy (SEM-FIB).

## 2. Materials and Methods

### 2.1. Raw powder

To perform this study, a commercial copper powder (Metco 55, Sulzer Metco, Kelsterbach, Germany) has been selected. This powder, already studied in the work of Guyot et al. [44], was prepared by the atomization method and exhibits micro-sized spherical particles. Its chemical purity is greater than 99.5%. The powder was sifted using stainless-steel sieves of 115 and 210 mesh and placed on a vibrating table for 45 min, in order to tighten its particle size distribution and thus have the most monodisperse particles possible. The structural and morphological properties of the powder were characterized using various techniques. The particle size distribution was determined using a laser diffraction particle size analyzer (Horiba LA-950V2, Kyoto, Japan). The measurements were performed in liquid form using distilled water as the suspending medium to disperse the microparticles (Appendix A). Moreover, powder density was determined by helium pycnometry, using a Micrometrics Accupyc II 1340 pycnometer (Nocross, GA, USA) at 23 °C.

The structural properties were analyzed by the X-ray diffraction (XRD) technique (Bruker D8 Advance with DAVINCI design, Karlsruhe, Germany) using Cu-Kα radiation (λ = 1.5406 Å). The XRD pattern acquisition conditions are: angular step of 0.02°, time per step of 0.3 s, angle range 20°–80°. The morphology of the copper particles was investigated using a scanning electron microscope (SEM) type LEO 1530VP (Oxford, UK) at an accelerating voltage of 10 kV. This SEM is equipped with an energy-dispersive X-ray spectroscopy (EDS) system, controlled by INCA Energy software, for composition analyses. Particles’ cross-sections were prepared by FIB, type Zeiss Crossbeam 550 (Oberkochen, Germany), to reveal their internal microstructure. X-ray Photoelectron Spectroscopy (XPS) was carried out using an “Axis Ultra DLD Kratos” spectrometer (Manchester, UK) in order to analyze the chemical bonds present on the surface of the particles. This technique provides valuable information about the oxidation state of copper. It is very sensitive to the composition of the extreme surface of the particles, since most of the signal (95%) comes from less than 10 nm deep from the surface. The XPS spectra were obtained with monochromatic Al Kα radiation (BE = 1486.71 eV, X-ray source operating at a power of 180 W). The pressure in the analysis chamber was maintained lower than 10^−9^ mbar during the spectral acquisition.

### 2.2. Reduction–Oxidation Treatments

Treatments of reduction–oxidation were applied to the raw powder in order to modify the thickness and nature of the native oxide layer at the surface of the copper particles. The objective is to master the initial insulating character of the powder bed and its sensitivity to electric and/or magnetic fields.

A reduction heat treatment was first performed to remove the native oxide layer at the surface of the copper particles. It was carried out by annealing the raw powder in a tubular furnace (type PYROX, Grenoble, France) at 400 °C during 2 h with a heating rate of 20 °C/min, under a reductive Ar-H_2_ atmosphere of high purity (2.7 vol % of H_2_) corresponding to an oxygen partial pressure of 10^−23^ atm. at 400 °C. The imposed flow rate was 10 mL/min. The oxygen partial pressure $(PO_{2})$ in the outlet gas of furnace is measured thanks to a potentiometric oxygen sensor (Micropoas sensor, Setnag, Marseille, France).

Then, different oxidation treatments were carried out by annealing the obtained reduced powder in a tubular furnace type CARBOLITE model “HVT 15/75/450” (Parsons Ln, UK). After a primary vacuum (≈50 Pa), the furnace chamber was filled with a gas mixture of 20 vol % O_2_ + 80 vol % N_2_ (dry air), ALPHAGAZ, Air Liquide, France, up to atmospheric pressure. Two oxidation temperatures were applied, i.e., 250 °C and 280 °C for 10 min [63], with heating and cooling rates of 10 °C/min, in order to develop oxide layers with various thicknesses and controlled microstructural characteristics. The gas flow was maintained at 0.145 L/min. During these treatments, a quantity of powder of 15 g was placed in an alumina crucible (L = 7 cm, W = 5 cm and H = 3 cm) in order to form a thin bed of copper powder with a thickness of about 1 mm. The limitation of the thickness of the powder bed aims at ensuring a homogeneous treatment of all the particles. Finally, to avoid any contamination related to the absorption of chemical species at the surface of copper particles and any further oxidation in air, both oxidized powders batches were transferred in a hermetic vessel and stored under Ar.

### 2.3. Electrical Measurements

The electrical properties of the raw and oxidized powders were carefully characterized. The powder was poured into a Teflon die with an inner diameter of 10 mm and inserted between two stainless-steel electrodes. The thickness of the powder bed was fixed at 0.30 mm. Before each electrical measurement, the whole sample–electrodes–die assembly was placed on a vibrating table during 5 min in order to prepare a controlled and reproducible stacking characteristics. Hence, it ensures a good reproducibility of experiments. This previous step leads to a low dispersion in the results. Moreover, this process allows to obtain a compactness of 62 ± 1% of the powder bed. Experiments were carried out at room temperature and under atmospheric pressure. They were performed using two mechanical configurations: a first one without applied load, in which the contact pressure between the electrodes and the powder is only due to the weight of the upper electrode (of approximately 5 N); a second one under uniaxial compaction, where the electrical responses of powder beds were in situ characterized by varying the applied stress from 0 to 300 MPa, using an Instron 5969 Universal Testing Machine (Norwood, MA, USA). The samples were compressed at a constant rate of 0.2 mm/min up to the desired stress using suitable load cells (capacity from 500 N to 50 kN).

The electrodes were connected to a source measure unit (SMU) type 2601A, Keithley Instr. Inc., Cleveland, OH, USA, with *Vmax* = 40 (V) and *Imax* = 1 (A); this instrument can simultaneously source and measure a DC current and/or voltage with high accuracy. A GPIB-type computer interface makes it possible to control the current–voltage source using a Labview code. The current–voltage characteristic of each sample was monitored by applying a scan of imposed DC current and measuring the induced voltage. The current was increased and decreased by step, with a time between each current step of 0.1 s and a total of 200 steps over the full applied current range. Finally, all the electrical measurements were repeated between 3 and 5 times to ensure a good reproducibility of the results.

## 3. Results and Discussion

### 3.1. Microstructural and Morphological Analyses

XRD analyses were performed to identify the crystalline phases present in the raw, reduced and oxidized Cu powders. The measured XRD patterns are presented in Figure 1. For the raw powder, copper (PDF 00-004-0836) and Cu_2_O (PDF 04-007-9767) oxide phases have been identified. According to the literature, the presence of a native oxide layer on the surface of copper particles is reported by several authors [64,65,66]. In contrast, no CuO oxide has been detected by XRD. However, XPS analysis confirms the presence of CuO at the extreme surface of the particles. This suggests that the copper particles of the raw powder are partially oxidized and thus have a native oxide layer on their surfaces, due to their exposure to air. Moreover, a low Tin dioxide (SnO_2_) content is revealed by XRD (PDF 04-003-0649) and confirmed by EDXS compositional analysis. Tin is known as a classic copper impurity.

For the reduced powder, at first sight, it may be considered that only the peaks characteristic of copper are indexed. However, by enlarging the low-angle part of the XRD diagram, the intensity of the characteristic Cu_2_O peak (2θ = 36.4°) is strongly attenuated after the reduction treatment at 400 °C for 2 h. Therefore, it seems that the applied heat treatment at 400 °C during 2 h in an Ar-H_2_ atmosphere allows to reduce strongly the native oxide layer at the surface of copper particles. The XPS analysis confirms the presence of a few amount of Cu_2_O oxide on the surface of reduced powders (results of XPS analysis are included in the Appendix A). Moreover, it is noticed that the reduction treatment at 400 °C for 2 h is not sufficient to completely remove the SnO_2_ oxide layer. In fact, the literature indicates a temperature higher than 400 °C for the reduction in SnO_2_ with hydrogen [67].

For both powders oxidized at 250 °C and 280 °C during 10 min, only the presence of the Cu_2_O oxide phase is detected, which is in agreement with [63,68,69]. In these oxidizing conditions, the Cu_2_O phase is highly orientated along the (111) plane. The CuO phase is not detected by XRD in oxidized powders, either because this phase is not crystallized or it is in a reduced quantity. XPS analyses confirm that oxidized Cu particles of both batches have traces of a few amount of CuO oxide on their extreme surface (an example of an XPS analysis is included in the Appendix A).

The morphology and the microstructure of the copper particles for the three powder batches were investigated by SEM (Figure 2). Figure 2A shows the surface morphology of the raw copper particles. The particles exhibit a spherical or quasi-spherical shape. At high magnification, their surfaces are rough due to the presence of the oxide layer. As the raw powder was sieved, the particle size distribution is tightened with a mean diameter (D_50_ in volume) of 87.5 µm, D_10_ = 69.8 µm and D_90_ = 113.3 µm. This contributes to the model character of the powder, as the particles within the powder beds for the electrical measurements are well quasi-monodisperse. The raw powder has a density of 8.82 ± 0.01 g/cm^3^. This value is slightly lower than the theoretical density of copper calculated by XRD refinement (8.903 ± 0.002 g/cm^3^) because of the presence of copper and tin oxides at the surfaces of the copper particles. Figure 2B shows the cross-section of the raw powder prepared by FIB. The different gray levels reflect the polycrystalline character of the Cu particles, with crystallites of approximately 7 µm in size. The FIB cross-section of the raw powder (Figure 2C) has been obtained at high magnification. It shows that particles are partially oxidized and have a thin native oxide layer of about 1 µm on their surfaces. It confirms the presence of a thin layer of oxide over the raw particles, as already suggested by both XRD and XPS experiments.

The surface morphology of the particles after the reduction treatment under hydrogen is shown in Figure 2D. A big change in surface morphology is found; the particles surface is smooth compared to the one of the raw powders. This indicates that the oxide layer on their surfaces has been largely removed, as suggested by the XRD analysis.

The FIB cross-sections of a Cu particle oxidized at 250 °C and 280 °C are shown in Figure 2E, F, respectively. One can notice that the oxide layer is distributed uniformly and homogeneously on the surface of the Cu particles after the oxidation treatment. The surface is quite rough, due to the germination and growth of the oxide layer. The thickness of the oxide layer for each powder batch was measured by SEM observations and is reported in Table 1.

### 3.2. Voltage/Current Characteristic: Quasi-Reversible Behavior at Low Current

Figure 3 shows the measured evolution of the voltage and the resistance for the raw powder submitted to an imposed current cycle of low intensity value (<10 µA). The contact between the particles is ensured by the weight of the upper electrode, which is approximately 5 N, and no external load is applied here. For a starting current of 0.1 µA, the raw powder exhibits a high initial resistance, around 20 MΩ. This value of R_0_ depends mainly on the geometry of the powder bed (i.e., thickness and diameter of the granular medium), the shaping process affecting the arrangement of the particles within the stack (compactness, coordination number, etc.) [70] and the electrical insulating character of the oxide layer on the surface of the copper particles. As shown in the previous paragraph, this layer is mainly composed of Cu_2_O. Its electrical properties depend on its morphology (especially the thickness, around 1200 nm for the raw powder) and its chemical composition. All these physicochemical characteristics of the copper-based granular medium determine the current percolation throughout the interparticles’ microcontacts.

As the electric current is increased further up to 10^−5^ A (I_max_), the voltage increases in a non-perfectly linear way up to 10 V (Figure 3A). Decreasing then the current from I_max_ to 10^−7^ A, one can notice that the *U*-*I* characteristic is not fully reversible and is marked by a short hysteresis. During this electrical cycle at low imposed current, a loss of resistance is observed, i.e., a reduction of 37% of the initial resistance (Figure 3B).

The microstructural modifications leading to this quasi-reversible behavior of the *R*-*I* curve at low current will be discussed further thanks to detailed SEM observations.

### 3.3. Irreversible Transition from an Insulating to a Conductive State

Figure 4 show the typical *U*-*I* and *R*-*I* characteristics measured by applying current cycles up to 1 A for the raw and oxidized Cu powders before and after mechanical shock. The *U*-*I* curves reveal several stages (indicated by arrows on Figure 4), which are associated with distinct electrical behaviors as discussed below.

***(a)*** 
**
*Raw powder*
**



**
*
1st cycle:
*
**


As mentioned above, at low current (up to 10 µA), a high electrical resistance, of the order of 10^8^ Ω, characterizes the raw powder. This is related to the existence of the insulating Cu_2_O layer of 1200 nm thick on the surface of the copper particles. As the electric current increases, the voltage increases quasi-linearly (*stage 1*) up to a maximum value of 15 V associated with a decrease in electrical resistance. When the current (I_0_) reaches about 0.4 mA, the voltage decreases suddenly by five orders of magnitude. It reveals a transition from an insulating to a conductive state (*stage 2*) where the resistance of the granular medium drops sharply for about eight decades, as shown in Figure 4B. Then, increasing further the current value up to 1 A, the voltage increases linearly again, with a stable resistance lower than 1 Ω (*stage 3*). When the current is then decreased again to 10 µA (*stage 4*), the voltage decreases linearly, showing a perfectly reversible behavior according to Ohm’s law.

Measurements of *U*-*I* curves were repeated five times for each powder batch following the same operating procedure. The obtained *U*-*I* curves follow exactly the same trajectories as the curves presented in Figure 4. In all cases, the breakdown voltage and the drop of the resistance were observed. The average values of initial resistance and electrical breakdown characteristics are reported in Table 2. A good reproducibility is noticed, especially for the breakdown voltage.

The variations of the initial resistance and breakdown voltage were evaluated by changing the thickness of the granular medium of the raw powder (see Table 3). Both properties increase linearly with the sample thickness in accordance with the increase in the number of layers within the granular stack and hence of the number of interparticles’ microcontacts. The conduction paths crossing the granular medium are therefore longer and more resistive. As the macroscopic breakdown voltage increases linearly with the sample thickness, it suggests that the microscopic breakdown voltage at the interfaces between adjacent particles remains relatively constant in all these experiments. These linear dependences show again the robustness and reproducibility of the applied experimental protocol.


**
*
2nd cycle:
*
**


Applying a second current cycle up to 1 A (directly after the first cycle and the induced transition), a perfectly linear and reversible *U*-*I* curve is obtained for the raw powder. The resistance of the granular medium remains low (below 1 Ω) and stable during this second cycle. The granular medium retains its conductive character.


**
*
After shock:
*
**


However, if a mechanical shock of low amplitude is applied to the raw powder, the granular medium recovers its original insulating state with a high resistance (10^8^ Ω). It shows that the previous process is not entirely reversible.

Applying a new current cycle, the obtained *U*-*I* curve is almost superimposed to the one measured during the first cycle. From 0.1 µA, the voltage across the granular compact first grows up to the breakdown voltage (15.5 V) where the powder becomes conductive. The drop of resistance is of the same order of magnitude (as shown in *stage 2* in Figure 4B). After the transition, increasing and decreasing the current gives rise to an ohmic dependence of the *U*-*I* curve.

***(b)*** 
**
*Oxidized powders*
**


The same electrical measurements were conducted for the powders oxidized at 250 °C and 280 °C (Figure 4). The initial resistances of both oxidized powders are three decades lower than the one of the raw powder (Table 2). As the particles’ morphology of the studied powders and their granular distribution are strictly identical as well as the shaping process applied before the electrical characterizations, it means that their microstructural characteristics (i.e., particle arrangement, compactness, coordination number, etc.) are very similar. Therefore, it suggests that this difference in initial resistance is mainly governed by the electrical properties of their respective oxide layers. Indeed, the chemical composition of the oxide layer could slightly vary because of the distinct atmosphere (i.e., partial pressure of oxygen) applied during the oxidation treatments used for the elaboration of the oxidized powders in comparison with the one encountered during the elaboration of the raw powder. Comparing now both oxidized powders, as expected, the powder oxidized at 250 °C is less resistive than the one treated at 280 °C. Its initial resistance is decreased by about 17%, which is strictly proportional to the reduction in the Cu_2_O thickness (185 nm vs. 220 nm respectively, see Table 1). This observation suggests that the intrinsic electrical resistivity of the oxide layers, and hence their chemical composition, is quite similar for both oxidized powders.

When the current increases (*stage 1*), the difference in resistance between both oxidized powders is maintained. To achieve the transition to a conductive state (*stage 2*), their electrical breakdown values are quite high (in voltage and current) in comparison with the ones measured for the raw powder (Table 2). Their breakdown voltage values (around 37 V) are increased twofold compared to the one of the raw powder (15 V). Hence, it is necessary to apply a higher current to obtain the transition for oxidized powders, despite their lower thicknesses of oxide layers. As discussed above when comparing their initial resistance values, this is probably attributed to difference in the chemical composition of their oxide layers (oxidized vs. raw powders). Comparing now both oxidized powders, the slight decrease in oxide thickness leads to a weak reduction in breakdown values.

After the electrical transition (*stages 3 and 4*), the down characteristic is linear, and the resistance remains low for both oxidized powders.

Finally, applying a second current cycle up to 1 A directly after the first cycle or after a mechanical shock, oxidized powders exhibit similar behaviors to the raw powder; these results are not displayed in Figure 4 for readability issues (the typical *U*-*I* characteristics for oxidized powders are included in the Appendix A).

### 3.4. Microstructural Evolutions within the Granular Medium and Their Chronology

In order to well understand the microstructural evolutions and their chronology occurring within the copper powder bed during the application of DC current cycles, careful SEM investigations were carried out at the different electrical stages.

***(a)*** 
**
*Microstructure after the quasi-reversible behavior at low current (stage 1)*
**


First, SEM observations were performed after a low current cycle (up to 10 µA for the raw powder) to analyze the microstructural modifications at the origin of the induced resistance decrease and non-perfectly reversible behavior. It reveals that the passage of electric current leads to the formation of “spots” at the microcontacts between the copper particles (Figure 5A). The size of these spots is approximately of a few micrometers in diameter (Figure 5B). In addition, as shown in Figure 5A, the majority of these spots are preferentially located along the perimeter of the contact area between two adjacent particles. This spatial distribution does not agree with the schematic representation of Falcon et al. [14], which assumes a random distribution of these spots at the microcontacts.

This observation can be explained by a higher current density along this perimeter because of the geometrical constriction of the current lines. Indeed, by equating the electrical contact between two adjacent particles to a constriction of radius *a* in a current tube as defined by Holm [71], it is shown that the current density along the contact radius (r) is given by:(1)$$J\left(r\right)=\frac{I}{2\pi a}\cdot \frac{1}{\sqrt{a^{2}-r^{2}}}$$

This spatial distribution suggests that the current flow is greater at the edges of the interparticle contact areas. Its integration shows that only half of the current flows through a disc of 0.866*a* diameter. Moreover, along this perimeter, it seems that the current flows non-uniformly and preferentially through these spots, which is probably because of local roughness effects and localized variations in thickness of the oxide layer. As the radius of these spots is approximately a few micrometers, even a current of low intensity passing through them is equivalent to a high current density (greater than a few tens Amps per mm^2^), which can eventually lead to melting by Joule effect. However, even without going to this extreme, the enlargement of the contact area by thermomechanical effect and the rapid raise in electrical conductivity of the semiconductor oxide layer (Cu_2_O) with temperature [63,72,73] are at the origin of the thermo-dependence of the contact resistance. As the current increases, local heating near these spots generates a decrease in local resistance. These spots represent hence the lowest electrical resistance path, and their formation leads to the measured reduction in the total electrical resistance.

***(b)*** 
**
*Microstructure after the electrical breakdown (stages 3–4)*
**


Figure 6 shows SEM images of the copper powder after application of a DC current cycle up to 1 A. After the transition from an insulating to a conductive state, clusters of welded copper particles are present in the granular medium (Figure 6A). These clusters are non-uniformly distributed. The particles in these clusters seem to have been weakly consolidated and welded. At high magnification within these clusters, micro-bridges can be noticed between adjacent particles. The size of these micro-bridges is approximately 10–15 µm (Figure 6B). At these micro-bridges, the oxide layer is no longer present. They correspond hence to metal-to-metal links. The localized sublimation of the oxide layer and the formation of these conduction paths allows the transition to the purely ohmic behavior observed in *stages 3–4*.

These observations agree with the theory of Falcon et al. [14] that characterized this electrical breakdown in a simple 1D-bead configuration. This transition comes from an electro-thermal coupling at the contact areas between the particles, leading to the formation of local microwelds by Joule effect [14]. The Kohlrausch equation [71] establishes this electro-thermal coupling:(2)$$T_{m}^{2}-T_{0}^{2}=\frac{U^{2}}{4L}$$
where *T_m_* is the maximum temperature reached at the microcontact, *T*_0_ the initial temperature, *L* is the Lorentz constant (2.45 × 10^−8^ V^2^/K^2^) and *U* is the local voltage. This relation allows estimating the local voltage at the microcontacts necessary to reach the sublimation temperature of the oxide layer (around 1800 °C [74]). Above a local voltage of 0.6 V, the contact temperature exceeds the sublimation point of the oxide layer, which leads to a metal-to-metal contact. Well-conductive electrical paths are hence created.

This approach coupling electrical and SEM characterizations gives new insight on the chronology of microstructural evolutions occurring within the granular medium under an applied DC current. It seems that the area of the “spots” formed during *stage 1* in the periphery of interparticles’ contacts areas increases with the passage of the electric current through them until the formation of microwelds between the particles after the destruction of the oxide layer. It promotes the formation of conductive electric paths, which ensures the transition from an insulating to a conductive state (*stage 2*). The presence of these metal-to-metal micro-bridges and the associated high conductivity of copper inhibit additional local heating by Joule effect. It explains the perfectly reversible *U*-*I* behavior during *stages 3 and 4*, where the final size of micro-bridges does not vary anymore with the current flow. Finally, as the size of these micro-bridges is relatively small, the consolidation of the granular medium is weak. Therefore, any mechanical shock can break them, leading back to the insulating state.

### 3.5. Pressure Effect on the Electrical Behavior

Electrical characterizations of the raw powder were also carried out when the powder bed is submitted to a uniaxial stress in a rigid die, as applied during SPS treatments. The purpose is here to analyze how the applied pressure alters the electrical response through modification of the microstructural characteristics (i.e., relative density, contact radius between adjacent particles, etc.).

**(a)** 
**
For low loads
**


For the low load tests (i.e., applied stress up to 40 MPa), the raw powder is poured in a Teflon die of 10 mm diameter with an initial sample thickness of 0.5 mm. Figure 7 shows the measured evolution of the electrical characteristics as a function of the applied stress. As the thickness of the granular stack decreases with the applied stress, the measured electrical responses are plotted and compared using relative electrical variables, i.e., electric field and current density instead of voltage and current, respectively. Whatever the applied stress, the electrical response is divided into four stages, as already observed during measurements without any applied load. When the current density increases, the electrical field increases. Then, beyond a critical value, a transition from an insulating to a conductive state occurs, and the resistance becomes very low (less than 1 Ω).

As shown in Table 4, the initial resistivity and the breakdown field decrease monotonously and strongly with the rise of the applied stress. These results agree with the ones obtained by Falcon et al. [53,75,76], who investigated the influence of applied loadings on the electrical transition occurring within a copper powder with a mean diameter of 100 µm. Here, these electrical properties are almost divided by three when the stress reaches 40 MPa. Their strong reduction is related to modifications of the microstructural features of the granular stacks [57,58,77]. The measured decrease in the relative density is one macroscopic physical expression of these microstructural changes. The applied pressure probably leads to an increase in the number of contact points between adjacent particles (i.e., coordination number). Therefore, the electric current can cross the granular compact more easily, which reduces the initial resistivity. In addition, the application of the pressure makes the contact surfaces between particles grow, which promotes the creation of microwelds during the flow of the electric current. This point will be discussed later based on SEM observations. As a result, the breakdown field of the medium decreases significantly. These experiments show how the DC Branly effect strongly depends on the pressure applied to the granular medium.

**(b)** 
**
For high loads
**


For the high load tests (i.e., stresses ranged from 100 to 300 MPa), the raw powder is poured in a stainless-steel die of 8 mm diameter, with an initial sample thickness of 1.0 mm. Figure 8A presents the measured curves of electric field vs. current density for various applied pressures. As noticed at low loads, the higher the applied pressure, the faster the electrical transition. The breakdown field falls to approximately 17.6 ± 3.5 kV/m under 300 MPa.

However, it can be noticed that for electrical measurements under very high loads, the electric field does not drop all at once when current density increases. Indeed, 80% of the tests at high loads present sudden and successive drops in electric field before reaching the conductive state. For instance, on the curve of the sample compacted at 200 MPa, four successive drops in electric field are detected. These drops are of various amplitudes, the first one being always the stronger. The first signs of this behavior appear from 40 MPa (Figure 7).

The same behavior is observed on the resistivity vs. electric field curve presented in Figure 8B. Indeed, the increase in the stress applied leads to successive drops in the resistivity of the granular medium before reaching the conductive state.

**(c)** 
**Microstructural observations**


Figure 9 provides a comparison of the surface morphologies of copper particles for increasing pressures applied on the granular stacks. At a low load of 30 MPa (Figure 9A), it appears that most particles remain almost spherical, with some evidences of plastic deformation on the surface of a few particles. The contact surfaces are circular in shape, with a diameter in the order of 12 µm. The increase in pressure at 200 MPa (Figure 9B) enlarges the contact surfaces. They remain circular, with diameters around 35 µm. However, inside them, some microcracks are detected in the oxide layer. At a higher stress of 300 MPa (Figure 9C), all the particles are faceted, and the contact surfaces are flat with a width of approximately 50 µm. Indeed, with the increase in applied pressure, the plastic deformation of copper particles becomes widespread, and the shape of the flattened faces turns into a polygon. At high magnification (Figure 9D), microcracking at the surface of the particles is generalized. The cracks are visible both inside the contact surfaces and on their periphery. These microcracks extend and widen with increasing pressure.

Based on these SEM observations, it seems that the decrease in the breakdown field with increasing applied pressure is correlated to an enhancement of contact surfaces between copper particles. Beyond 40 MPa, this surface increase is associated with microcracking of the oxide layer on the surface of the copper particles. The local damage of this insulating layer promotes the formation of conductive paths around these microcracks. This local phenomenon could explain the electrical transition characterized by successive drops in electric field (Figure 7 and Figure 8) before a stable and generalized conductive transition.

### 3.6. Discussion

Deep investigations of electrical response of copper granular media have been carried out here. The effects of pre-oxidation state and mechanical loading have been analyzed on the Branly effect induced at relatively small applied DC currents. Correlations between measured electrical behaviors and fine SEM characterizations of the microstructure gives new insight on the involved conduction mechanisms and their chronology.

Falcon et al. [52,53] previously characterized the electrical properties of a granular medium composed of similar commercial copper particles under low applied uniaxial stress (up to 20 MPa). They noticed similar electrical behavior, which was characterized by a non-linear decrease in the resistance before a “sudden” transition from an insulating to a conductive state beyond a critical value. Whatever the applied stress, they noticed that the transition threshold always corresponds to the same dissipated power. This suggests that the transition probably originates from thermal instability. They make the hypothesis that this transition is related to electro-thermal processes occurring at interparticle microcontacts, which leads to electric breakdown of oxide layers on metal particles and to the formation of microwelds. However, no microstructural evidence of these microcontacts formation has been provided.

In the present work, by conducting electrical characterizations at low current and without applied external load combined to post-mortem SEM observations, it is shown that before the transition, the observed quasi-reversible behavior and the measured reduction in the resistance is in fact related to the formation of spots at the microcontacts between the copper particles. These spots are preferentially located along the perimeter of the contact areas, which agrees with the spatial current density distribution estimated by considering the constriction effect in a current tube as defined by Holm. The formation of these spots at low current corresponds to the preliminary stage before the ignition of transition. The spots surfaces grow with increasing current by electro-thermal coupling, until the sudden electrical transition to the conductive state. As shown, this transition is characterized by the destruction of the insulating oxide layer at the microcontacts between some particles and the formation of some particles clusters (non-uniformly distributed). Microwelds are formed within these clusters, creating well-conductive electrical paths. The presence of these metal-to-metal micro-bridges and the associated high conductivity of copper inhibit additional local heating by Joule effect. It explains the perfectly reversible *U*-*I* behavior after transition.

Furthermore, through controlled reduction–oxidation treatments, the measurements carried out here show how the chemical composition and the thickness of the oxide layer directly affect its electrical resistivity and consequently the electric breakdown by Joule effect leading to the electrical transition of the copper granular medium.

When a mechanical stress is applied to the granular medium, the DC Branly effect is strongly affected. As previously mentioned by Falcon et al. [53], at low loads (up to 40 MPa), the breakdown field of the electrical transition decreases with increasing pressure. As shown here, this reduction is correlated to microstructural modifications, especially the increase in relative density and contact surfaces between adjacent particles.

Moreover, the investigation of pressure effect has been extended to high applied stresses, up to a few hundred MPa compatible with the pressures encountered in SPS sintering. For applied pressure higher than 40 MPa, successive drops in electric field are detected during the electrical transition. These successive drops are induced by microcracking of the insulating oxide layer, which favors the formation of local conductive paths. The highlight of this local damage phenomenon of the oxide layer shows that the Branly effect is favored at high loads.

## 4. Conclusions

This work has been devoted to the study of the Branly effect occurring within copper granular medium under an applied DC current. This effect corresponds to an electric transition from an insulating to a conductive state under increasing current. The powder beds are composed of spherical copper particles with a mean diameter of 87.5 µm. Suitable reduction–oxidation treatments were applied to control the pre-oxidation state of these copper particles, especially the thickness of the oxide layer on their surfaces (ranged here between 185 and 1200 nm). Moreover, the DC Branly effect has been investigated by varying the pressure applied to the granular medium. The measured electrical responses were correlated to careful SEM observations. It allows a thorough understanding of the microstructural evolutions and their chronology at the origin of this transition.

At low current, the reduction in the resistance is induced by the formation of spots, preferentially located along the perimeter of the contact areas between particles due to higher current density. When the current increases, these spots enlarge until the electrical breakdown of the insulating oxide layer at the microcontacts between some particles. It leads to the electrical transition marked by microwelds, around ten µm in diameter.

The application of a mechanical stress strongly affects the DC Branly effect. At low stress (up to 40 MPa), the relative density increases and the copper particles begin to deform plastically, enhancing the contact surfaces between adjacent particles. The measured breakdown field decreases rapidly, almost by a factor of 3. For applied stress higher than 40 MPa, microcracking of the oxide layer is observed. It induced a discontinuous electrical transition, marked by successive drops in electric field. Consequently, it is shown that the Branly effect and the associated formation of local metal-to-metal bridges are favored at high loads.

These investigations will help to understand the electro-thermo-mechanical coupled behavior of powder beds during electric current-assisted sintering (ECAS) treatments, especially in the SPS technique. The Branly effect probably occurs during the early stages of the densification process. The in-depth understanding provided here of the electro-mechanical conditions and the microstructural evolutions leading to this electrical transition will allow to define the operating parameters favoring the formation of these microwelds between particles and enhancing hence the sintering kinetics.

## Figures and Tables

**Figure 1 materials-15-04096-f001:**
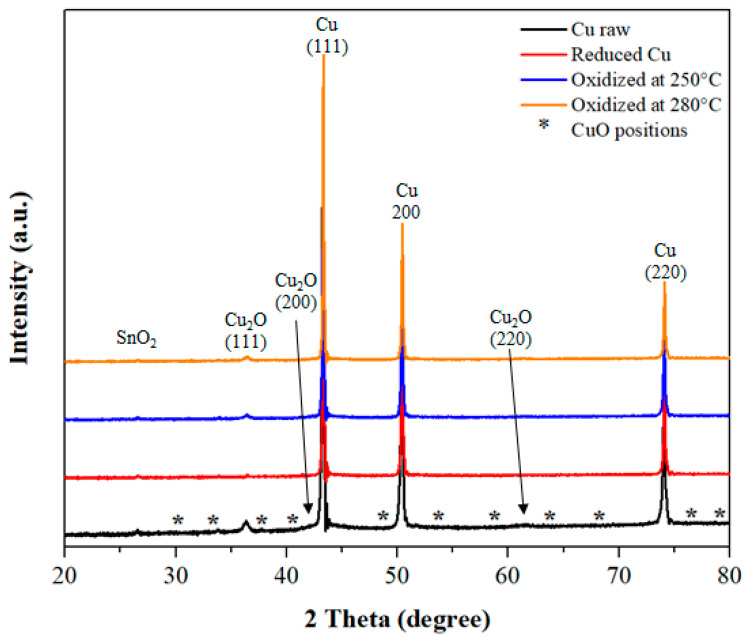
XRD patterns of raw, reduced and oxidized Cu powders.

**Figure 2 materials-15-04096-f002:**
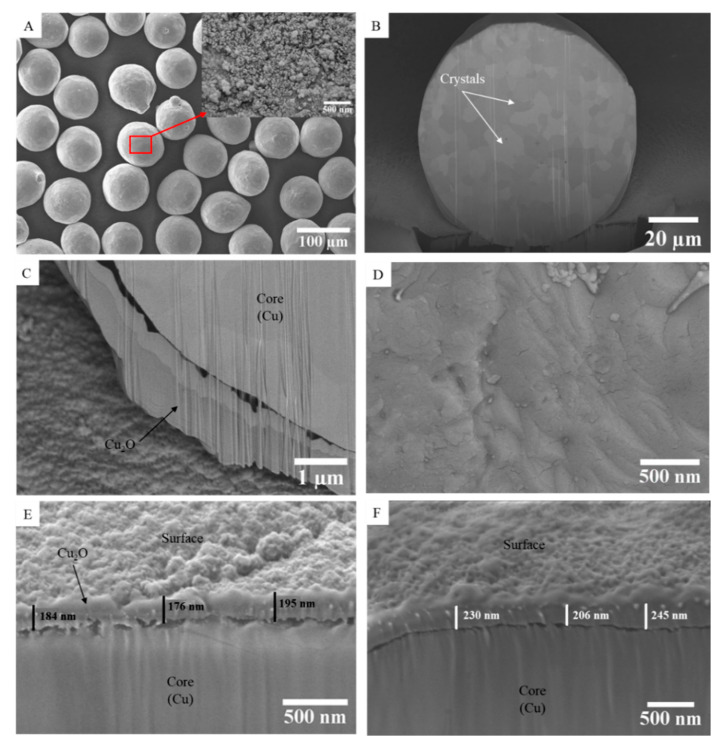
SEM observations of the Cu powders: (**A**) raw powder at low magnification; (**B**) cross-section of one particle of the raw powder prepared by FIB; (**C**) cross-section of one particle of raw powder; (**D**) surface morphology of the reduced powder; (**E**,**F**) cross-sections of one particle of powder oxidized at 250 °C and 280 °C, respectively.

**Figure 3 materials-15-04096-f003:**
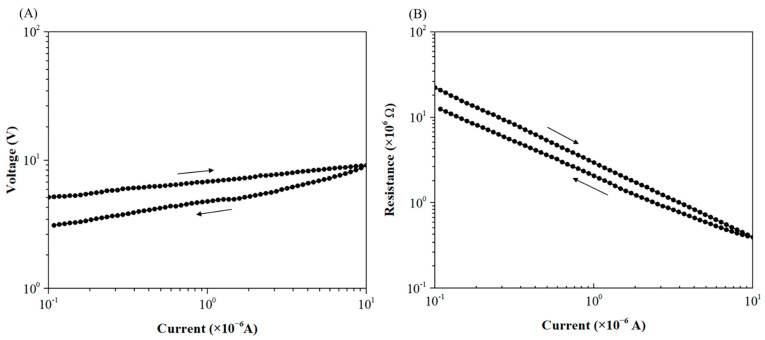
Typical *U*-*I* (**A**) and *R*-*I* (**B**) characteristics (logarithmic scale) for the raw powder at low imposed current.

**Figure 4 materials-15-04096-f004:**
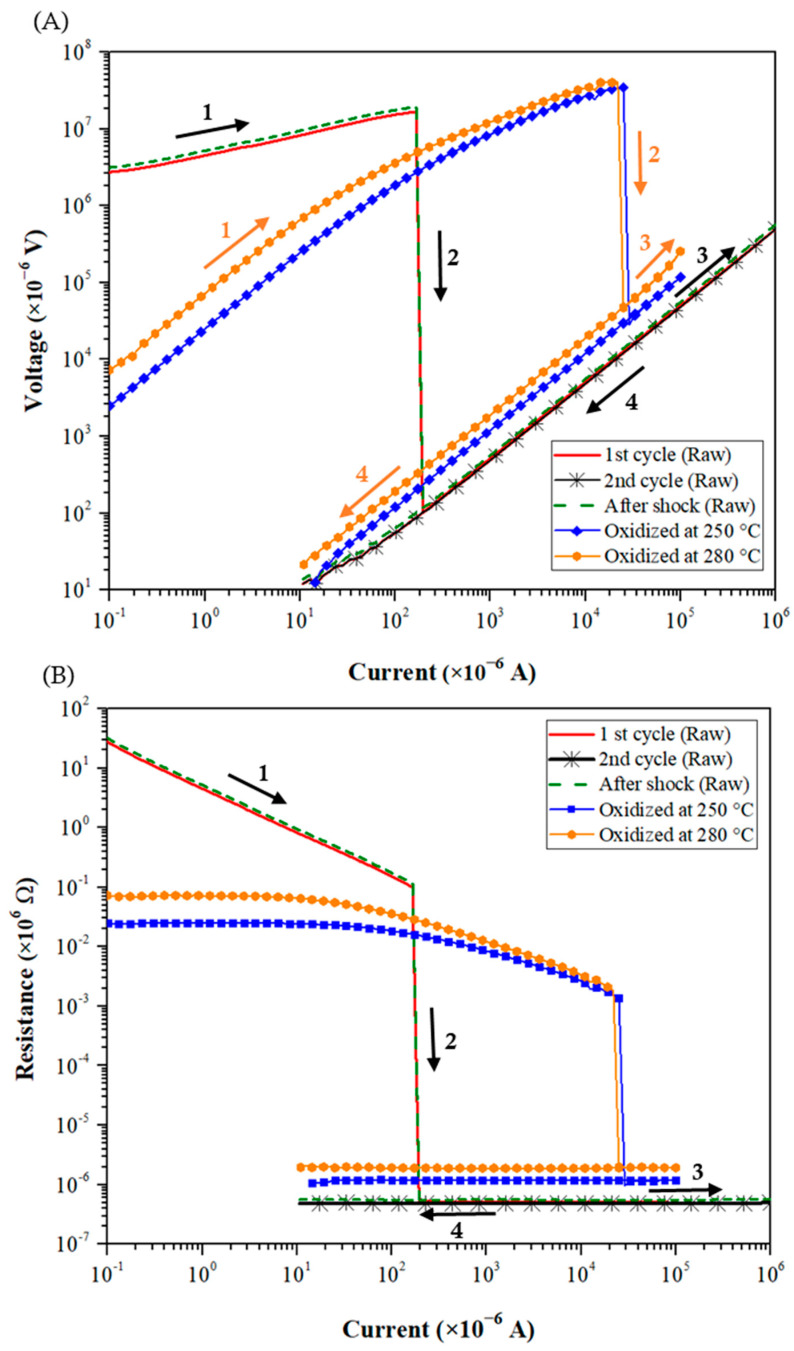
Typical *U*-*I* (**A**) and *R*-*I* (**B**) characteristics (logarithmic scale) measured during current cycles up to 1 A for the raw and oxidized Cu powders before and after mechanical shock.

**Figure 5 materials-15-04096-f005:**
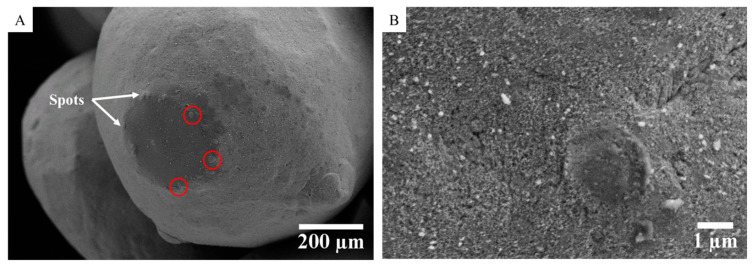
SEM observations of the raw powder after application of a low DC current up to 10 µA (*stage 1*): (**A**) spots (encircled in red) located along the perimeter of the contact area between two adjacent particles; (**B**) high magnification of these spots.

**Figure 6 materials-15-04096-f006:**
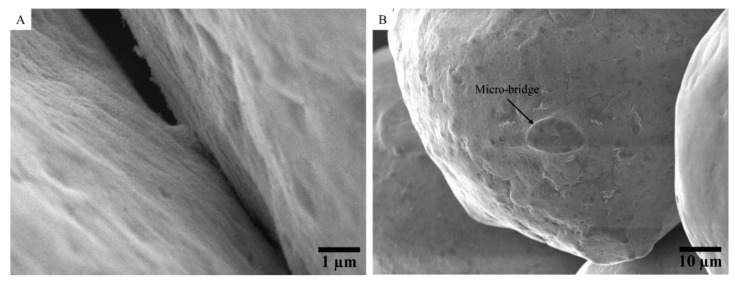
SEM observations of the raw powder after application of a DC current up to 1 A (*stages 3−4*): (**A**) clusters of copper particles; (**B**) micro-bridges at interparticles’ contacts areas.

**Figure 7 materials-15-04096-f007:**
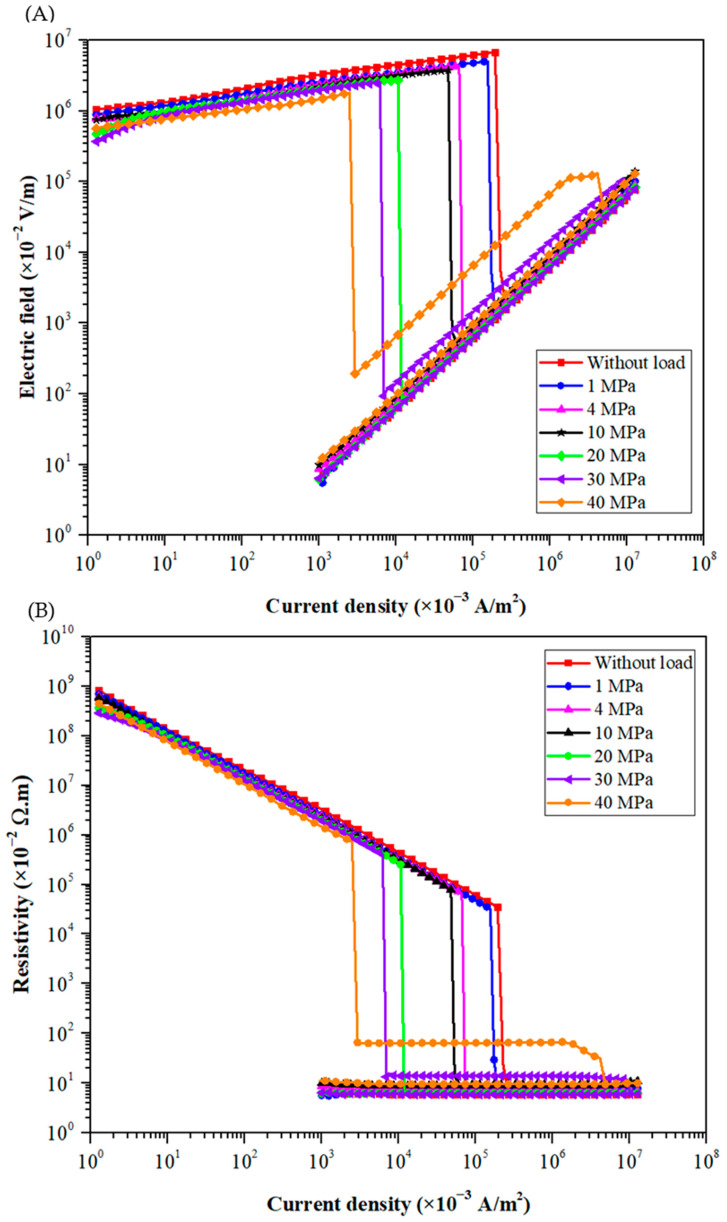
Evolutions of (**A**) electric field vs. current density and (**B**) resistivity vs. current density during current cycles up to 1 A for the raw powder submitted to a low uniaxial stress up to 40 MPa.

**Figure 8 materials-15-04096-f008:**
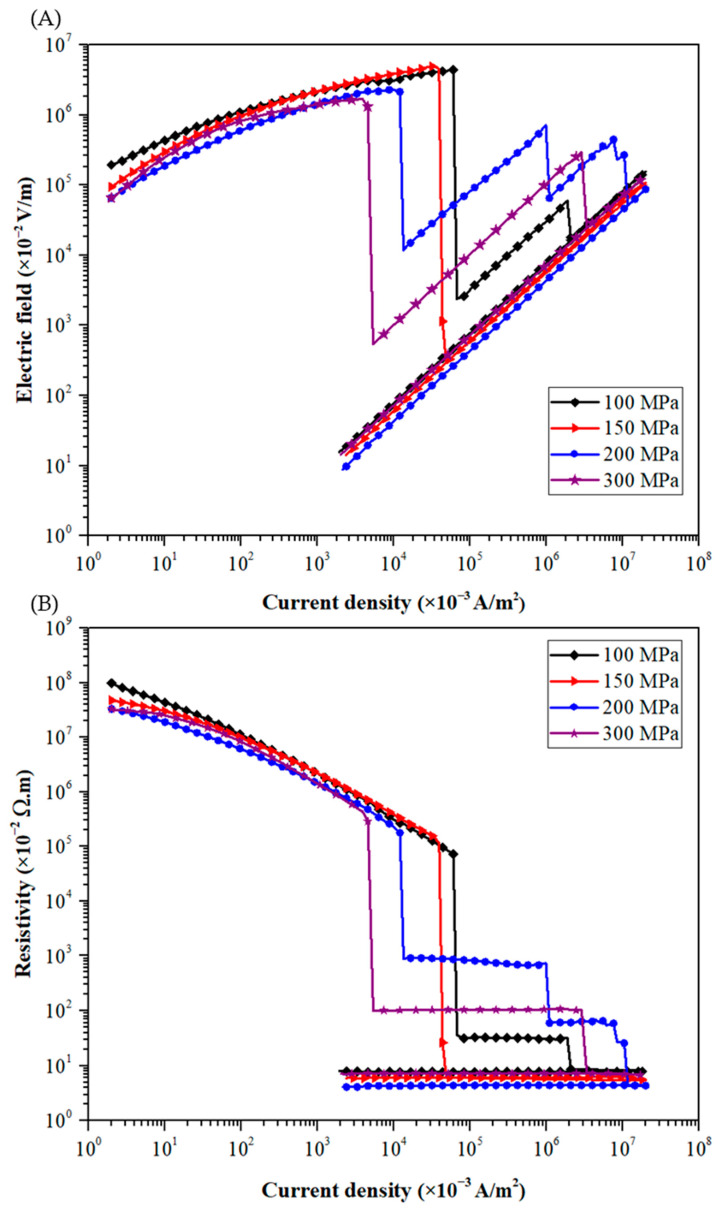
Evolutions of (**A**) electric field vs. current density and (**B**) resistivity vs. current density during current cycles up to 1 A for the raw powder submitted to a high uniaxial stress (ranged from 100 to 300 MPa).

**Figure 9 materials-15-04096-f009:**
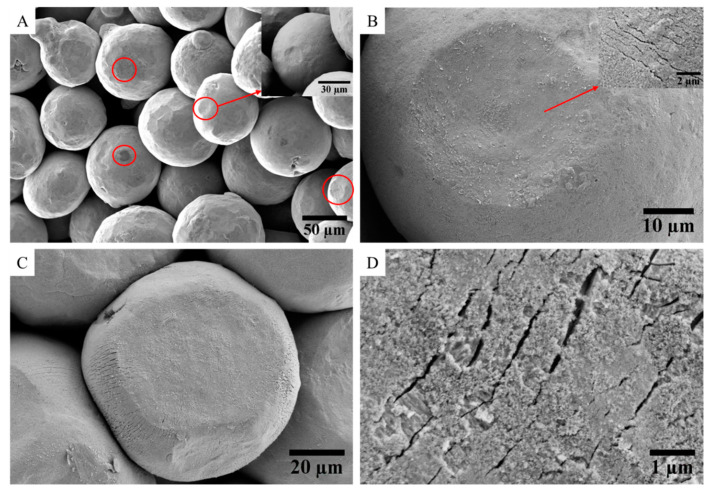
Microstructures of granular compacts obtained under different uniaxial stresses: (**A**) at 30 MPa; (**B**) at 200 MPa; (**C**,**D**) at 300 MPa.

**Table 1 materials-15-04096-t001:** Thickness of the oxide layer for each powder batch.

Powder Batch	Thickness (nm)
Raw	1200 ± 100
Oxidized at 280 °C	220 ± 30
Oxidized at 250 °C	185 ± 25

**Table 2 materials-15-04096-t002:** Initial resistance and electrical breakdown characteristics for the raw and oxidized powders. The thickness of the samples was 0.30 mm.

	Initial Resistance (MΩ)	Breakdown Current (mA)	Breakdown Voltage (V)
Oxidized at 250 °C	0.042 ± 0.030	23.0 ± 3.5	36.5 ± 3.6
Oxidized at 280 °C	0.051 ± 0.020	25.4 ± 4.9	37.0 ± 3.5
Raw	22 ± 5	0.32 ± 0.1	15.6 ± 2.3

**Table 3 materials-15-04096-t003:** Dependence of the initial resistance and breakdown voltage to the thickness of the granular medium for the raw powder. Each measurement was repeated five times.

Sample Thickness (mm)	Initial Resistance (MΩ)	Breakdown Voltage (V)
0.50 ± 0.01	52 ± 8	33.4 ± 3.5
0.40 ± 0.01	35 ± 10	24.3 ± 4.3
0.30 ± 0.01	22 ± 5	15.6 ± 2.3

**Table 4 materials-15-04096-t004:** Dependence of the relative density, initial resistivity and breakdown field on the pressure applied to the raw powder. Each measurement was repeated five times.

Stress (MPa)	Relative Density (%)	Initial Resistivity (MΩ.m)	Breakdown Field (kV/m)
No load	51.7 ± 2.0	387 ± 55	56.6 ± 4.5
1	54.3 ± 1.5	334 ± 28	39.0 ± 5.2
4	56.0 ± 3.5	274 ± 54	34.3 ± 5.1
10	58.5 ± 3.0	239 ± 63	32.4 ± 5.1
20	60.2 ± 1.3	171 ± 16	27.3 ± 3.8
30	63.1 ± 4.0	130 ± 17	24.0 ± 5.3
40	67.1 ± 3.9	115 ± 7	21.7 ± 4.0

## Data Availability

Not applicable.

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
