# Peer review of "Investigation of Electrical Transitions in the First Steps of Spark Plasma Sintering: Effects of Pre-Oxidation and Mechanical Loading within Copper Granular Media"

_materials, 2022, doi:10.3390/ma15124096_

Round 1

Reviewer 1 Report

The author present a thoroughly study on the electrical behaviour of copper under the first step of SPS process. Different thickness of oxide layers on the material play crucial role to reflect the electric breakdown field which response to the transition from insulating to conductive states. This study is interesting to the researcher working on the field assisted sintering process. However, some issues in the manuscript might need to be addressed in order to publish in the journal.

In page 7 line 289-290, under the hypothesis that the oxide phases are fully crystalized. Author might need to examine the CuO and Cu2O phase under XPS to see whether there is any oxide phase in the surface.  From the XRD figure, I still can observed very small amount of Cu2O and SnO2 phases in reduced Cu patterns. So I would not say the reduction can completely remove the oxide phases.

In page 8 line 310,  theoretical density of copper (8.96 g.cm-3 ), Authors are suggested to use XRD refinement to calculated the theoretical density of the copper and compare to the measured density.

Author Response

Reviewer 01:

Point 1: In page 7 line 289-290, under the hypothesis that the oxide phases are fully crystalized. Author might need to examine the CuO and Cu2O phase under XPS to see whether there is any oxide phase in the surface. From the XRD figure, I still can observe very small amount of Cu2O and SnO2 phases in reduced Cu patterns. So I would not say the reduction can completely remove the oxide phases.

Response 1: We have considered your proposal to perform an XPS analysis on the reduced powder. The XPS analysis indicates the presence of a very small amount of Cu2O oxide on the surface of the powders. So, to be as clear as possible, we have modified the paragraph in the manuscript to indicate that the oxides are strongly reduced but not completely removed, as below:

Results and discussion: lines 285-295 pages 7-9: “For the reduced powder, at first sight it may be considered that only the peaks characteristic of copper are indexed. But by enlarging the low-angle part of the XRD diagram, the intensity of the characteristic Cu2O peak (2θ=36.4°) is strongly attenuated after the reduction treatment at 400 °C for 2h. Therefore, it seems that the applied heat treatment at 400 °C during 2 hours in an Ar-H2 atmosphere allows to reduce strongly the native oxide layer at the surface of copper particles. The XPS analysis confirms the presence of a few amount of Cu2O oxide on the surface of reduced powders (results of XPS analysis are included in the supplementary material). Moreover, it is noticed that the reduction treatment at 400 °C for 2h is not sufficient to completely remove the SnO2 oxide layer. In fact, the literature indicates a temperature higher than 400 °C for the reduction of SnO2 with Hydrogen [67]”.

Point 2: In page 8 line 310, theoretical density of copper (8.96 g.cm-3), Authors are suggested to use XRD refinement to calculated the theoretical density of the copper and compare to the measured density.

Response 2: The density calculation based on XRD refinement analysis (8.903 ± 0.002 g/cm3) provide value close to the measured density (8.82 ± 0.01 g/cm3). We have included this calculation in the manuscript:

Results and discussion: lines 313-316 page 8: “The raw powder has a density of 8.82 ± 0.01 g/cm3. This value is slightly lower than the theoretical density of copper calculated by XRD refinement (8.903 ± 0.002 g/cm3) because of the presence of copper and tin oxides at the surfaces of the copper particles”.

Reviewer 2 Report

Dear Authors,

I find Your work very interesting and worth publication, although I would like to support your article with a few minor suggestions.

Minor suggestions

1.      [line 11] – SPS is now used not only in science, but also in some industrial applications. It is surely  a promising method, but is became conventional and has been popularized over last 10 years, especially in DC case.

2.      Please reformat bibliography to a form [11-15], when it is possible to it.

3.      [line 56] please separate Amperes in brackets propertly with spaces, we use space after coma, ie (0 , 750, 850, 1000 A)

4.      Please unify °C units:

4.1.   [lines 226-228, 291, 391 and others] 400° C -> 400 °C

Celsius degree is a unit, and it can not be separated. °C – correct

Writing 600° C is interpreted as (600 angle degrees) Coulombs (electrical charge unit), and it does not correspond with any subject within article.

4.2.   What does the dot “.” mean in  “C.min-1”, I suppose You wanted to write Cmin-1 or K/min? I’m not a purist of SI units when describing technological parameters, but consider presenting rates in Kelvin/time unit. The same notation takes place in g.cm3. I supposed You meas multiplication sign × (alt + 0215)?

4.3.   [line 431] – although there are two correct ways of writing °C unit (100°C or 100 °C), please do not use both randomly in one article. As I see separated form is preferred

Author Response

Reviewer 02:

Point 1: SPS is now used not only in science, but also in some industrial applications. It is surely a promising method, but is became conventional and has been popularized over last 10 years, especially in DC case.

Response 1: We have changed the beginnings of the abstract and of the introduction to indicate that the SPS sintering process is a conventional method, such as the following:

Abstract – line 11 page 1: “Spark Plasma Sintering (SPS) becomes a conventional and promising sintering method for powder consolidation”.

Introduction – line 30 page 1: “Spark Plasma Sintering (SPS), which belongs to a set called Field Assisted Sintering Technique (FAST) [1-2], is used for the synthesis, consolidation and assembly of metals [3-5], ceramics [6-7] and composites materials [8-10]”.

Point 2: Please reformat bibliography to a form [11-15], when it is possible to it.

Response 2: All bibliographic references in the manuscript have been formulated according to the format you have indicated.

Point 3: [line 56] please separate Amperes in brackets properly with spaces, we use space after coma, ie (0, 750, 850, 1000 A)

Response 3: Sorry for this mistake. We have separated Amperes in brackets properly with spaces.

Point 4.1 & 4.3: Please unify °C units

Response 4.1 & 4.3: We have modified all the units (°C) to be in correct and uniform form in our manuscript.

Point 4.2: What does the dot “.” mean in “C.min−1 ”, I suppose You wanted to write Cmin or K/min? I’m not a purist of SI units when describing technological parameters, but consider presenting rates in Kelvin/time unit. The same notation takes place in g.cm3. I supposed you means multiplication sign × (alt + 0215)?

Response 4.2: Yes, it means the multiplication sign × (alt + 0215). However, all the notations C. min-1 and g.cm3 have been changed to the format C/min and g/cm3.

Reviewer 3 Report

referee report 

materials-1751666-peer-review-v1

Investigation of electrical transitions in the first steps of spark 2 plasma sintering: effects of pre-oxidation 

and mechanical 3 loading within copper granular media

Anis Aliouat et al.

In the present manuscript, the authors discuss electrical transitions during spark-plasma sintering (SPS) of Cu

granular media. As SPS is a very important technique to prepare dense ceramic materials, this detailed investigation

is clearly of importance for a better understanding of the SPS process itself. Thus, this is a very interesting 

topic and so well suited for publication in Materials.

The manuscript comprises 9 figures, 4 tables and 76 references are included, which provide a good overview on 

the field.

The manuscript is well prepared and well arranged, with all required experimental details given in the

Experimental Section (2/2.1/2.2/2.3).

However, there are several technical points which require to be addressed prior to publication:

# Please define abbreviations used in the abstract also within the abstract.

# In all the graphs (Fig.3/Fig.4/Fig.7/Fig.8), please include the multiplication factor (10^-6 or so) into the 

   unit. The resulting graph will look much better and easier to be understood.

# Physical quantities should be written in italics everywhere.

# Please format all chemical formulae properly, also in the article titles in the reference list.

# Reference list:

   If there are article numbers, there is no need for "p."

   In case of beginning and end page, it would be "pp.", but the journal style does not include this.

To summarize, the present manuscript presents very interesting material and is well prepared. Thus, I can recommend

the manuscript to be suitable for publication after a minor revision.

Author Response

Reviewer 03:

Point 1: Please define abbreviations used in the abstract also within the abstract.

Response 1: We have defined the abbreviations used in the abstract.

Point 2: In all the graphs (Fig.3/Fig.4/Fig.7/Fig.8), please include the multiplication factor (10^-6 or so) into the unit. The resulting graph will look much better and easier to be understood.

Response 2: The graph of figures (Fig.3/Fig.4/Fig.7/Fig.8) has been modified to respect the format you have recommended.

Point 3: Physical quantities should be written in italics everywhere.

Response 3: We have updated the physical quantities in italic format throughout the manuscript.

Point 4: Please format all chemical formulae properly, also in the article titles in the reference list.

Response 4: Sorry for this mistake. All the chemical formulae have been corrected in the manuscript.

Point 5: Reference list: If there are article numbers, there is no need for “p.” In case of beginning and end page, it would be "pp.", but the journal style does not include this.

Response 5: All bibliographic references in the manuscript have been formulated by removing “p” while respecting the format indicated in the journal.
